# British Columbia's Experience after Implementation of the *Treponema pallidum* Reverse Algorithm and PCR Detection, 2015 to 2020

Muhammad Morshed,[a,c] Min-Kuang Lee,[a] Jonathan Laley,[a] Darrel Cook,[b] Annie Mak,[a] Navdeep Chahil,[a] Venessa Ryan,[b] Carolyn Montgomery,[b] Sylvia Makaroff,[b] Sarah Malleson,[b] Barbra Arnold,[b] Troy Grennan,[b,d] Jason Wong,[b] Mel Krajden[a,c]

aBritish Columbia Centre for Disease Control Public Health Laboratory, Vancouver, British Columbia, Canada
bClinical Prevention Services, BC Centre for Disease Control, Vancouver, British Columbia, Canada
cDepartment of Pathology and Laboratory Medicine, University of British Columbia, Vancouver, British Columbia, Canada
dDepartment of Medicine, University of British Columbia, Vancouver, British Columbia, Canada

**ABSTRACT** British Columbia (BC) implemented the syphilis reverse screening algorithm and *Treponema pallidum* PCR testing in 2014. We summarize the performance characteristics of the algorithm, together with PCR direct detection, and report on syphilis cases identified from 2015 to 2020. Prior to 2015, samples for syphilis diagnosis were first screened by rapid plasma reagin (RPR). As of 2015, sera were screened by the Siemens Advia Centaur syphilis assay (enzyme immunoassay [EIA]). Positive and equivocal samples were reflex tested by a *T. pallidum* passive particle agglutination assay (TPPA) and RPR. We used *T. pallidum* DNA PCR on clinical samples and restriction fragment length polymorphism analysis to identify azithromycin resistance mutations. Case/epidemiological data were obtained from the BC surveillance system. Of 1,631,519 samples screened by the EIA, 72,492 (4.4%) were positive and 187 (<0.1%) were equivocal. Of EIA-positive/equivocal samples, 10.6% were false positive, and false positivity was higher at lower EIA indices. The reverse algorithm detected 4,693 late latent syphilis cases that likely would have been missed by RPR screening. PCR had a very high sensitivity of 100% versus 52.9% and 52.4% for dark-field (DF) and immunofluorescence (IF) microscopy, respectively. The azithromycin resistance mutation A2058G was identified in 96% of PCR-positive samples, and A2059G was identified in 4%. Annually, there were 944 to 1,467 syphilis cases, with 62% in men who reported male sexual partners. The reverse algorithm had a low false-positive rate and very few equivocal screening results but did identify previously undiagnosed late latent syphilis cases. PCR was more sensitive than both DF and IF microscopy for direct diagnosis and enabled monitoring for azithromycin resistance.

**IMPORTANCE** In this study, we summarize the performance characteristics of the algorithm, together with PCR direct detection and epidemiological analysis, and report on syphilis cases identified from 2015 to 2020. This allowed us to paint a complete picture of the outcome of the utilization of the reverse algorithm for diagnosing syphilis cases. The study clearly showed that the reverse algorithm had a low false-positive rate and very few equivocal screening results but did identify previously undiagnosed late latent syphilis cases. PCR was more sensitive than both DF and IF microscopy for direct diagnosis and enabled monitoring for azithromycin resistance.

**KEYWORDS** PCR, reverse algorithm, screening, syphilis

Syphilis, caused by *Treponema pallidum* subsp. *pallidum*, is reemerging worldwide (1), with rising case numbers observed in both Canada (2) and the United States (3). In British Columbia (BC), syphilis cases have been rising since 2010, predominantly among gay, bisexual, and other men who have sex with men (MSM) (4). Syphilis infection is

Address correspondence to Muhammad Morshed, Muhammad.morshed@bccdc.ca.

The authors declare no conflict of interest.

categorized into three main stages, primary, secondary, and latent; staging of infections is based on laboratory results together with clinical assessment. Primary syphilis usually presents as a spontaneously resolving painless ulcer. Secondary syphilis occurs after systemic dissemination of *T. pallidum*, typically with a widespread maculopapular rash and other nonspecific systemic symptoms, and it also resolves spontaneously. Latent syphilis is asymptomatic, but if it becomes symptomatic (tertiary stage), an involvement of virtually any organ is possible, including the central nervous and cardiovascular systems. Neurosyphilis may occur at any stage of the disease (5).

Since *T. pallidum* cannot easily be cultured, serologic tests, including rapid plasma reagin (RPR) tests and *T. pallidum* enzyme immunoassays (EIAs), are the mainstays of routine diagnosis (5, 6). The traditional algorithm uses a nontreponemal test (usually RPR) for screening, with positive screen tests confirmed by *T. pallidum*-specific tests. The reverse algorithm, first described in the 1990s (7), involves screening by a *T. pallidum*-specific assay with reflex testing by a second *T. pallidum*-specific assay to confirm the EIA screen results, followed by the RPR test to assess for active infection. Both the RPR test and the EIA usually become positive about 2 to 4 weeks after infection, which may result in false-negative serology during early infection, especially for nontreponemal tests (5, 8). However, direct detection tests on fluid from ulcerative lesions, regional lymph nodes, or other infected tissues can enable early-phase case detection, which may help to mitigate transmission by identifying acute treatable infections quickly. Dark-field (DF) and immunofluorescence (IF) microscopy have been used historically for direct detection (9, 10), while more recently, nucleic acid amplification by PCR has been shown to have higher sensitivity (11–14). Of note, while *T. pallidum* subsp. *pallidum* is the most common species in developed nations, other *T. pallidum* subspecies exist and differ in their pathogenicity, but there is >95% DNA homology, and all are indistinguishable by serologic testing (15).

In September 2014, the BC Centre for Disease Control (BCCDC) Public Health Laboratory switched from RPR- to EIA-based screening (i.e., the reverse algorithm) for syphilis diagnosis. In this paper, we retrospectively describe the performance characteristics of the reverse algorithm and PCR-based *T. pallidum* DNA detection from 2015 to 2020. We also report the detection of *T. pallidum* azithromycin resistance in DNA-positive samples and syphilis case data for the same period.

## RESULTS

**Syphilis serology algorithm.** For the transition to the reverse algorithm laboratory validation set of 1,067 samples with known results by the traditional algorithm, the EIA was positive for 29/29 positive (relative sensitivity versus RPR, 100%; 95% confidence interval [CI], 85.0 to 100%) and 1/1,038 negative (relative specificity, 99.9%; 95% CI, 99.5 to 100%) samples.

From 2015 to 2020, the proportion of positive and equivocal EIA results averaged 4.5% (Table 1 and Fig. 1), and only a very small proportion (0.01%) were equivocal. The reflex *T. pallidum* passive particle agglutination assay (TPPA) was positive for 89.1% of samples, and the corresponding RPRs were positive for 37.9%. Those who had previously tested TPPA positive were not retested.

Overall, 10.6% (7,689/72,679) of EIAs with signals higher than 0.9 were deemed to be false positive, i.e., EIA positive or equivocal/TPPA negative/RPR negative. For EIA signals of 0.9 to 1.09 (equivocal), 1.10 to 9.99 (positive), and ≥10.0 (positive), 78.1% (147/187), 52.4% (6,750/12,890), and 1.3% (793/59,602), respectively, were false positive (Fig. 2). The false-positivity rate declined with increasing EIA signal strength. Of EIA equivocal samples (index, 0.9 to 1.09), reflex testing with the TPPA and RPR demonstrated that about 12% are likely to be true syphilis infections.

Of note, the EIA versus RPR for initial screening resulted in the identification of 4,693 EIA- and TPPA-positive/RPR-negative latent syphilis cases, which were confirmed by chart review. Most of these likely would not have been diagnosed using the traditional algorithm (Fig. 1).

**TABLE 1** British Columbia syphilis reverse algorithm screening and confirmation test results, 2015 to 2020[c]

| Yr | No. of samples screened | No. of EIA-positive samples | No. of EIA-equivocal samples | Total no. (%) of positive/equivocal samples | Samples tested by TPPA | | | | Samples tested by RPR[a] | |
|---|---|---|---|---|---|---|---|---|---|---|
| | | | | | No. positive | No. previously positive[b] | Total no. (%) positive | No. (%) negative | No. (%) positive | No. (%) negative |
| 2015 | 235,420 | 9,377 | 52 | 9,429 (4.0) | 1,598 | 6,609 | 8,207 (87.0) | 1,222 (13.0) | 4,361 (46.3) | 5,068 (53.7) |
| 2016 | 254,589 | 10,471 | 28 | 10,499 (4.1) | 1,429 | 7,748 | 9,177 (87.4) | 1,322 (12.6) | 4,263 (40.6) | 6,236 (59.4) |
| 2017 | 264,696 | 11,054 | 31 | 11,085 (4.2) | 1,522 | 8,422 | 9,944 (89.7) | 1,141 (10.3) | 4,224 (38.1) | 6,861 (61.9) |
| 2018 | 284,023 | 12,989 | 29 | 13,018 (4.6) | 1,766 | 9,819 | 11,585 (89.0) | 1,433 (11.0) | 4,593 (35.3) | 8,425 (64.7) |
| 2019 | 312,870 | 15,463 | 29 | 15,492 (5.0) | 2,055 | 11,920 | 13,975 (90.2) | 1,517 (9.8) | 5,596 (36.1) | 9,896 (63.9) |
| 2020 | 279,931 | 13,138 | 18 | 13,156 (4.7) | 1,713 | 10,136 | 11,849 (90.1) | 1,307 (9.9) | 4,525 (34.4) | 8,631 (65.6) |
| Total | 1,631,529 | 72,492 | 187 | 72,679 (4.5) | 10,083 | 54,654 | 64,737 (89.1) | 7,942 (10.9) | 27,562 (37.9) | 45,117 (62.1) |

[a]Samples testing positive for the first time were considered active cases; if the RPR titer was ≥4-fold higher than that of the previously RPR-tested sample, the case was also considered active. TPPA-positive, RPR-negative cases were considered latent.
[b]Patients who tested TPPA positive on a previous sample were not retested.
[c]EIA, enzyme immunoassay screen test; TPPA, treponemal passive particle agglutination assay; RPR, rapid plasma reagin test.

**T. pallidum direct detection.** Of all suspected syphilis lesion samples subjected to direct detection tests, 814 were tested by two or more of the DF, IF, and PCR tests. Table 2 shows the sensitivity, specificity, positive and negative predictive values, and false-positive and -negative rates for each test. Overall, PCR demonstrated 99.5% sensitivity, while those of DF and IF microscopy were lower (52.9% and 52.4%, respectively). All three tests had high specificity and positive predictive values, but negative predictive values were lower for both DF and IF microscopy, which resulted in a high false-negative rate for these tests.

**Azithromycin resistance.** Table 3 shows the *T. pallidum* azithromycin resistance mutations detected among 220 PCR-positive samples. Overall, 95.8% of those tested had the A2058G mutation, while 4.2% had A2059G. A2059G was not detected in BC until 2019.

**Syphilis cases.** Syphilis case numbers during the study period ranged from 944 to 1,467 annually (Table 4). The proportions of cases by stage of infection were similar for all years, except for an almost doubling of secondary cases (10.6% to 18.4%). Males

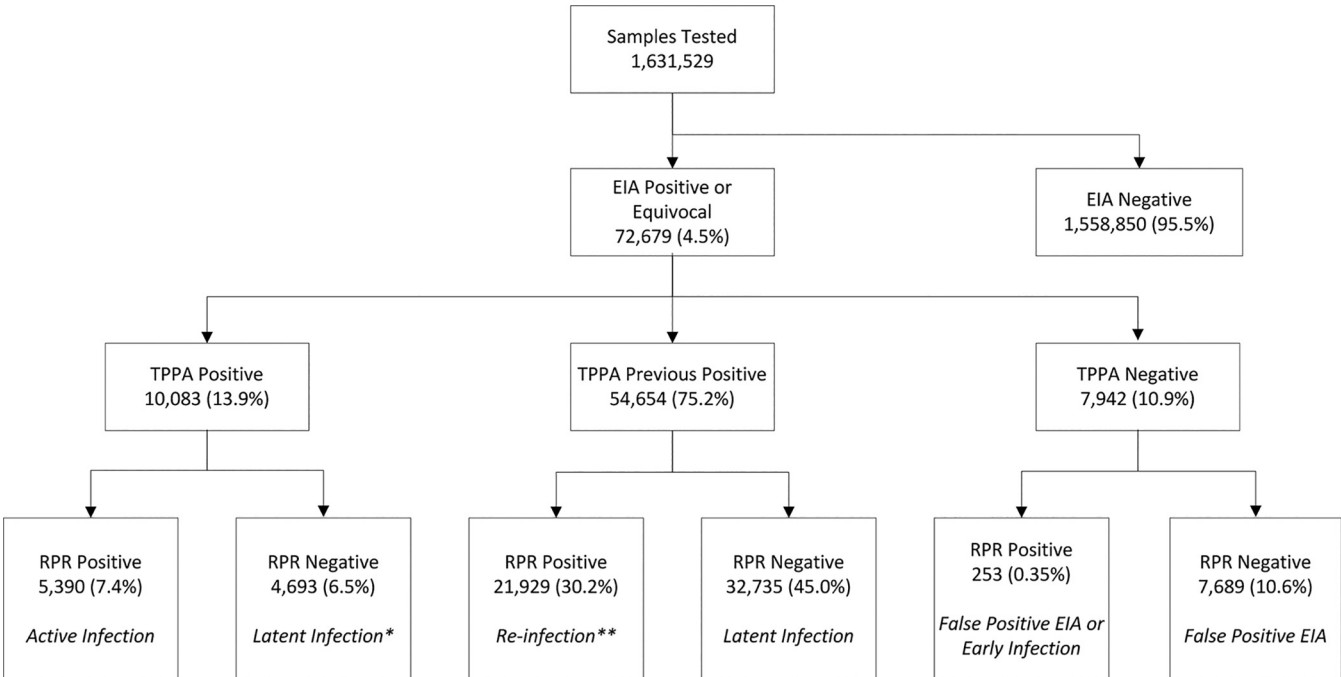

**FIG 1** Syphilis serology reverse screening algorithm, BCCDC Public Health Laboratory, 2015 to 2020. *, these latent infections would likely have been undetected and not treated when using rapid plasma reagin (RPR) as a screen test; **, the RPR titer is ≥4-fold higher than that of a previous specimen. EIA, enzyme immunoassay screen test; TPPA, treponemal passive particle agglutination assay.

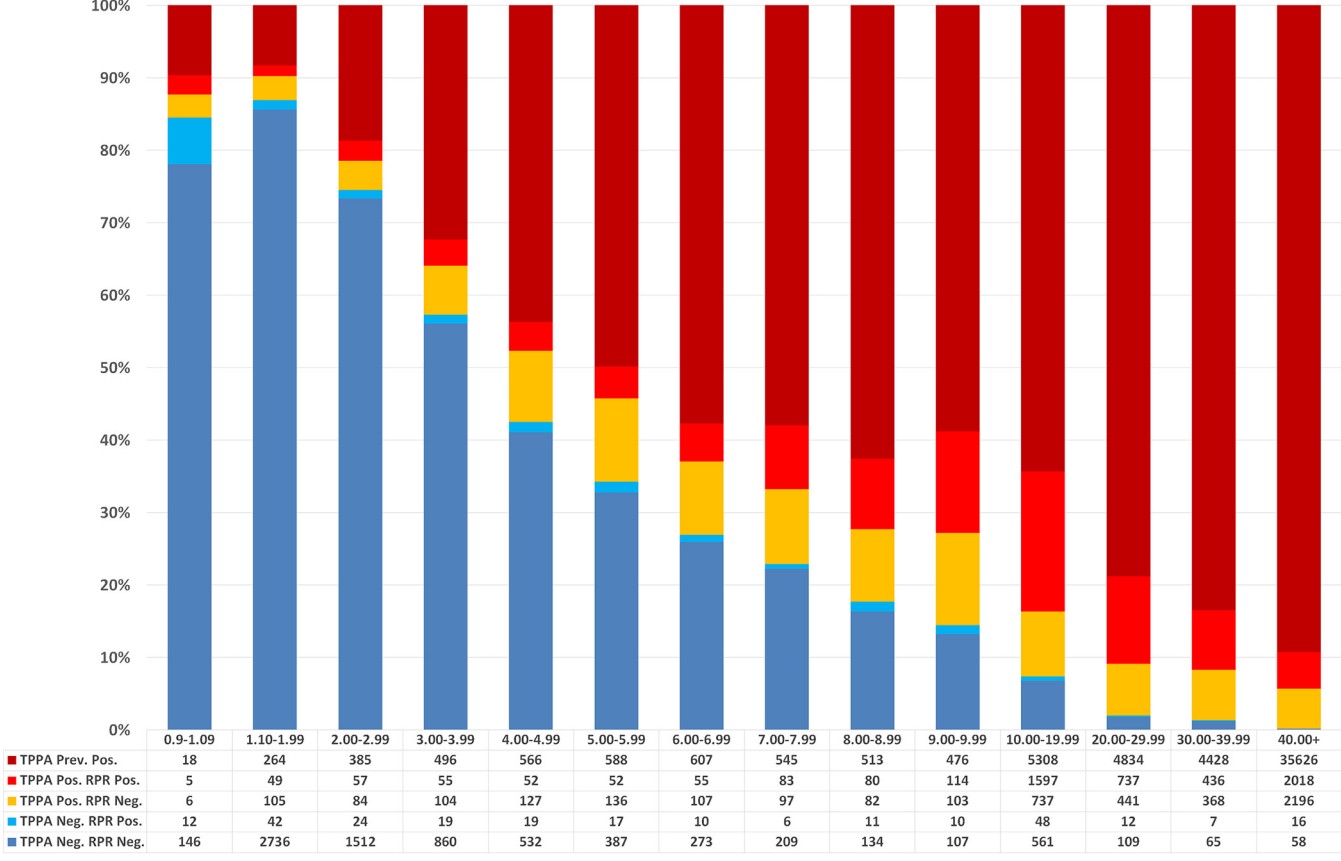

| | 0.9-1.09 | 1.10-1.99 | 2.00-2.99 | 3.00-3.99 | 4.00-4.99 | 5.00-5.99 | 6.00-6.99 | 7.00-7.99 | 8.00-8.99 | 9.00-9.99 | 10.00-19.99 | 20.00-29.99 | 30.00-39.99 | 40.00+ |
|---|---|---|---|---|---|---|---|---|---|---|---|---|---|---|
| ■ TPPA Prev. Pos. | 18 | 264 | 385 | 496 | 566 | 588 | 607 | 545 | 513 | 476 | 5308 | 4834 | 4428 | 35626 |
| ■ TPPA Pos. RPR Pos. | 5 | 49 | 57 | 55 | 52 | 52 | 55 | 83 | 80 | 114 | 1597 | 737 | 436 | 2018 |
| ■ TPPA Pos. RPR Neg. | 6 | 105 | 84 | 104 | 127 | 136 | 107 | 97 | 82 | 103 | 737 | 441 | 368 | 2196 |
| ■ TPPA Neg. RPR Pos. | 12 | 42 | 24 | 19 | 19 | 17 | 10 | 6 | 11 | 10 | 48 | 12 | 7 | 16 |
| ■ TPPA Neg. RPR Neg. | 146 | 2736 | 1512 | 860 | 532 | 387 | 273 | 209 | 134 | 107 | 561 | 109 | 65 | 58 |

**FIG 2** TPPA and RPR results stratified by syphilis EIA signal index, 2015 to 2020. Note that false-positive result classification is based on interpretation for an individual sample; some of these may represent early infection or treated cases from the remote past.

accounted for 85.8% of cases (Table 5), many of whom reported male sexual partners (61.9%). We also observed moderate increases over time among males who report only female partners (13.4% to 19.6%) and females who report only male partners (42.2% to 60.4%). Of concern, there were seven congenital syphilis cases identified from 2019 to 2020.

## DISCUSSION

The high EIA sensitivity and specificity that we reported are consistent with reports by others (5, 8, 16). An important feature of the reverse algorithm is its ability to detect latent syphilis cases that would be missed if RPR is used for screening, i.e., EIA and TPPA positive but RPR negative (8, 17). We identified a large number of latent cases, which presents an opportunity for treatment to reduce the potential long-term sequelae. Of note, those not treated in the past can be identified since BC maintains a database of all syphilis diagnoses and treatments that have occurred since the 1960s. Of the 4,693 EIA-positive, RPR-negative cases identified in the present study, follow-up by

**TABLE 2** Performance characteristics of syphilis direct detection tests, 2015 to 2020 (*n* = 814)[a]

| Test | Total no. of samples | No. of positive samples | No. of negative samples | Sensitivity (%) (95% CI) | Specificity (%) (95% CI) | Positive predictive value (%) (95% CI) | Negative predictive value (%) (95% CI) | False-positive rate (%) | False-negative rate (%) |
|---|---|---|---|---|---|---|---|---|---|
| DF | 40 | 9 | 31 | 52.9 (28.5, 76.1) | 100 (82.2, 100) | 100 (62.9, 100) | 74.2 (55.1, 87.5) | 0 | 47.1 |
| IF | 790 | 104 | 686 | 52.4 (45.0, 59.6) | 99.2 (98.0, 99.7) | 95.2 (88.6, 98.2) | 86.9 (84.1, 89.3) | 0.8 | 47.6 |
| PCR | 809 | 190 | 619 | 99.5 (96.7, 100) | 100 (99.2, 100) | 100 (97.5, 100) | 99.8 (99.0, 100) | 0 | 0.5 |

[a]DF, dark-field microscopy; IF, immunofluorescence microscopy.

**TABLE 3** British Columbia *Treponema pallidum* azithromycin resistance detection, 2015 to 2020

| Yr | No. of samples tested | No. (%) of samples with mutation detected | No. (%) of samples with A2058G | No. (%) of samples with A2059G | No. (%) of samples with no mutation detected |
|---|---|---|---|---|---|
| 2015 | 16 | 15 (93.8) | 15 (100) | 0 | 1 (6.3) |
| 2016 | 18 | 17 (94.4) | 17 (100) | 0 | 1 (5.6) |
| 2017 | 15 | 15 (100) | 15 (100) | 0 | 0 |
| 2018 | 16 | 16 (100) | 16 (100) | 0 | 0 |
| 2019 | 75 | 72 (96.0) | 68 (94.4) | 4 (5.6) | 3 (4.0) |
| 2020 | 51 | 51 (100) | 48 (94.1) | 3 (5.9) | 0 |
| Total | 220 | 215 (97.7) | 206 (95.8) | 9 (4.2) | 5 (2.3) |

BCCDC clinic physicians suggested that approximately 1,000 of these individuals may not have received treatment in the past. It is possible that some of these infections may have resolved spontaneously or as a result of individuals undergoing antibiotic treatments for other indications.

Since the EIA usually remains positive for life in infected individuals, the same is not the case for RPR, which usually reverts to negative following treatment and may also revert to negative even in untreated individuals (5). Thus, EIA screening in the reverse algorithm will have a higher screening positivity rate than RPR, leading to more samples requiring confirmatory testing. The additional cost is offset by the benefit of increased detection of late latent cases that would be missed by RPR screening. As a result, the cost-effectivenesses of the two approaches for routine screening are almost equivalent (18). However, one U.S. study reported that for screening high-prevalence populations such as MSM, initial screening by the EIA is inefficient because of high exposure rates in the past and that the traditional algorithm with RPR is preferred (19). However, this approach would be difficult to implement in a large program such as the one in BC, which routinely screens high volumes of samples from multiple different subpopulations, and sexual risk information does not usually accompany the specimens.

We demonstrated that PCR testing is more sensitive than DF and IF tests for syphilis direct detection, consistent with other reports (14, 20, 21). DF positivity requires a high bacterial burden, while PCR amplification increases sensitivity for smaller numbers of organisms. Identifying and treating these active infections earlier could mitigate onward syphilis transmission, especially among difficult-to-reach populations. However, the routine use of direct detection tests is challenging due to the difficulty of obtaining optimal samples from early cases. Lesions may not be visible if they occur inside the anal or vaginal cavities, and in later stages of infection, the lesions may have resolved. DF microscopy is time sensitive due to the loss of *T. pallidum* motility within about 20 min (5), thus requiring microscopy equipment and trained personnel either at the clinic or in a nearby

**TABLE 4** British Columbia confirmed syphilis cases by stage of infection, 2015 to 2020[a]

| | No. (%) of cases by stage of infection | | | | | | | | |
| | Infectious | | | | Noninfectious | | Congenital | | |
| Yr | Primary | Secondary | Early latent | Probable early latent | Late latent | Tertiary | Early | Late | Annual total no. of cases |
|---|---|---|---|---|---|---|---|---|---|
| 2015 | 151 (13.2) | 121 (10.6) | 335 (29.4) | 153 (13.4) | 381 (33.4) | 0 | 0 | 0 | 1,141 |
| 2016 | 128 (12.8) | 152 (15.2) | 310 (31.0) | 169 (16.9) | 240 (24.0) | 1 (0.1) | 0 | 0 | 1,000 |
| 2017 | 117 (12.4) | 111 (11.8) | 286 (30.3) | 176 (18.6) | 254 (27.0) | 0 | 0 | 0 | 944 |
| 2018 | 173 (14.3) | 157 (13.0) | 428 (35.5) | 162 (13.4) | 287 (23.8) | 0 | 0 | 0 | 1,207 |
| 2019 | 192 (13.1) | 210 (14.3) | 484 (33.0) | 180 (12.3) | 395 (26.9) | 1 (0.1) | 3 (0.2) | 2 (0.1) | 1,467 |
| 2020 | 136 (11.2) | 223 (18.4) | 396 (32.6) | 157 (12.9) | 297 (24.5) | 2 (0.2) | 2 (0.2) | 0 | 1,213 |
| Total | 897 (12.9) | 974 (14.0) | 2,239 (32.1) | 997 (14.3) | 1,854 (26.6) | 4 (0.1) | 5 (0.1) | 2 (0.0) | 6,972 |

[a]Note that BC syphilis case definitions are available at http://www.bccdc.ca/health-professionals/clinical-resources/case-definitions/syphilis.

**TABLE 5** British Columbia confirmed syphilis cases by reported sexual orientation, 2015 to 2020[a]

| Yr | Gender | No. (%) of cases by sexual orientation | | | | | Total no. of cases (% of annual total) | Annual total no. of cases |
|----|--------|----------|--------------|------------|----------------------------|------------------|------------------------|-----------|
| | | Bisexual | Heterosexual | Homosexual | Partner is transgender[b] | Unknown/ missing | | |
| 2015 | Male | 46 (4.6) | 135 (13.4) | 631 (62.8) | | 192 (19.1) | 1,004 (88.0) | 1,141 |
| | Female | | 57 (42.2) | | | 78 (57.8) | 135 (11.8) | |
| | Transgender | | | 1 (50.0) | | 1 (50.0) | 2 (0.2) | |
| 2016 | Male | 39 (4.4) | 110 (12.4) | 572 (64.7) | | 163 (18.4) | 884 (88.5) | 999 |
| | Female | | 47 (41.2) | | | 67 (58.8) | 114 (11.4) | |
| | Transgender | | | 1 (100) | | | 1 (0.1) | |
| 2017 | Male | 39 (4.9) | 90 (11.2) | 512 (63.8) | | 161 (20.1) | 802 (85.0) | 943 |
| | Female | 1 (0.7) | 53 (38.7) | | | 83 (60.6) | 137 (14.5) | |
| | Transgender | | 1 (25.0) | 2 (50.0) | | 1 (25.0) | 4 (0.4) | |
| 2018 | Male | 36 (3.5) | 92 (8.9) | 659 (63.4) | 4 (0.4) | 248 (23.9) | 1,039 (86.8) | 1,197 |
| | Female | 1 (0.7) | 58 (39.7) | | | 87 (59.6) | 146 (12.2) | |
| | Transgender | 1 (8.3) | | | | 11 (91.7) | 12 (1.0) | |
| 2019 | Male | 63 (5.1) | 150 (12.0) | 770 (61.8) | 4 (0.3) | 259 (20.8) | 1,246 (85.6) | 1,456 |
| | Female | 1 (0.5) | 109 (52.9) | | | 96 (46.6) | 206 (14.1) | |
| | Transgender | | | | | 4 (100) | 4 (0.3) | |
| 2020 | Male | 41 (4.2) | 193 (19.6) | 543 (55.2) | 2 (0.2) | 204 (20.8) | 983 (81.2) | 1,211 |
| | Female | 4 (1.8) | 136 (60.4) | | | 85 (37.8) | 225 (18.6) | |
| | Transgender | | | | | 3 (100) | 3 (0.2) | |
| Total | Male | 264 (4.4) | 770 (12.9) | 3,687 (61.9) | 10 (0.2) | 1,227 (20.6) | 5,958 (85.8) | 6,947 |
| | Female | 7 (0.7) | 460 (47.8) | | | 496 (51.5) | 963 (13.9) | |
| | Transgender | 1 (3.8) | 1 (3.8) | 4 (15.4) | | 20 (76.9) | 26 (0.4) | |

[a]Note that cases where gender was unspecified ($n = 25$) are not included in this table.
[b]Collection of "transgender" as a gender of the sexual partner commenced in 2018.

laboratory. Therefore, DF microscopy is not feasible in most settings. While PCR testing had higher sensitivity, the longer turnaround time to the receipt of results may require a repeat clinic visit to initiate treatment, while theoretically, DF results might be available while the client is still at the clinic. Another potential solution would be the use of point-of-care rapid antigen or nucleic acid amplification tests.

Macrolides, including azithromycin, have been used in the past to treat infectious syphilis (22), but increasing resistance has been reported for syphilis and other pathogens (23–25). Since *T. pallidum* is not easily cultivable, restriction fragment length polymorphism (RFLP) analysis of PCR-positive samples can be used to detect resistance mutations. The first azithromycin-resistant syphilis cases were reported in San Francisco, CA, in 2003 (26) and later in other parts of the world (27). BC began monitoring for azithromycin resistance following a mass treatment initiative in the 1990s (22), and monitoring has continued until the present time, in part to determine whether resistance detected in the years following might decline over time. From 2000 to 2003 in BC, only 1 of 47 PCR-positive cases demonstrated resistance, and it was a travel-related heterosexually acquired case. By 2004, four of nine tested cases showed resistance, all among males who report only male sexual partners (28), and by 2015 to 2020, almost all of the tested cases were resistant. The majority displayed the A2058G mutation, but A2059G has also been detected recently. Fortunately, no penicillin-resistant syphilis cases have been reported in the literature to date, but studies have demonstrated *T. pallidum* penicillin resistance mutations that could potentially lead to clinically important resistance (29). Thus, there may be a future need to monitor penicillin resistance, especially given the rising syphilis rates in Canada and elsewhere.

From 2015 to 2020, males accounted for the majority of syphilis cases in BC, consistent with the situation elsewhere in the world (1), although we have seen recent increased

trends among females who report only male sexual partners as well as among males who report only female partners. Of concern is the recent identification of congenital cases, which may reflect an increasing prevalence among females and a failure to access health care services or testing among those who are pregnant. One-time prenatal screening in the first trimester or at the first prenatal visit is recommended but not mandatory in BC. Since 2019, when congenital cases began to be detected, it is also recommended at delivery.

A strength of this study is the large number of patient samples that have been screened and confirmed using the reverse algorithm and the increased detection of latent syphilis. However, we acknowledge some limitations. EIAs classified as falsely positive could be an indication of early infection, and these need to be followed up with repeat serology to confirm whether they represent true infections or false-positive screen tests. We also recognize that the sensitivities of direct detection tests might be underestimated as we required RPR positivity rather than EIA or RPR positivity to confirm cases with discordant results for the three direct tests. In addition, low-volume samples from newborns may affect serologic diagnosis since it may not be possible to perform the full algorithm, but correlation with maternal results is usually helpful for interpretation and for selecting the most important tests to perform on the newborn.

In conclusion, we demonstrated the successful implementation of the reverse screening algorithm, which automated the initial screening process and increased the detection of latent syphilis cases compared to the traditional algorithm. We were able to treat more than 1,000 of these previously undetected late latent cases. Due to its high sensitivity, PCR testing likely increased the detection of active early syphilis and enabled the laboratory to undertake surveillance for azithromycin resistance.

## MATERIALS AND METHODS

**Syphilis reverse serologic screening algorithm.** Prior to the implementation of the reverse algorithm, 1,067 samples for routine RPR screening (29 confirmed syphilis positive and 1,038 negative) were retested by the EIA to validate the transition.

Following the implementation of the algorithm, serum samples for syphilis diagnostic testing were screened by the Advia Centaur syphilis assay (Siemens Healthcare Diagnostics, Inc., Newark, NJ) (EIA), which produces a signal index with a range of 0.0 to 45.0; an index of ≥1.1 is positive, and an index of 0.9 to 1.1 is equivocal. EIA-negative samples are not tested further, while EIA-positive and -equivocal samples are reflex tested by the qualitative *Treponema pallidum* passive particle agglutination assay (TPPA; Fujirebio, Inc., Malvern, PA) and the quantitative rapid plasma reagin (RPR) test (BD RPR card test; Becton, Dickinson and Company, Franklin Lakes, NJ). The TPPA is not repeated if the patient has previously tested TPPA positive. For RPR, samples are screened at a 1:1 dilution, and positive samples are titrated to determine the endpoint. Cerebrospinal fluid (CSF) samples are tested by the BD BBL venereal diseases reference laboratory (VDRL) test (Becton, Dickinson and Company, Franklin Lakes, NJ). Samples are screened at a 1:1 dilution, and positive samples are titrated to determine the endpoint.

An active syphilis diagnosis is based on serologic results and is either (i) newly EIA, TPPA, and RPR positive or (ii) EIA and TPPA positive with an RPR titer ≥4-fold higher than that of a previous sample. Cases where both the EIA and TPPA are positive and the RPR is negative can have a number of interpretations but are generally either latent infections or, less commonly, early infections (30). The full case definition is in the supplemental material. For the present analysis, EIA-positive and TPPA- and RPR-negative results are considered EIA false positives, although these may also represent very early cases or those treated in the remote past. We also assessed false-positivity rates based on the EIA signal strength index together with combined RPR and TPPA results.

**T. pallidum direct detection.** DF and IF testing were performed as described in *A Manual of Tests for Syphilis* (31). Suspected lesions were cleaned with moist gauze, and the expressed exudate was applied to two glass slides. Slides for DF microscopy were wet mounted with a coverslip and examined by dark-field microscopy. For IF microscopy, the smears were air dried, fixed in acetone for 5 min, allowed to air dry, and stained with fluorescein isothiocyanate (FITC)-labeled rabbit anti-*Treponema pallidum* antibodies (Meridian Life Science, Inc., Memphis, TN). After staining, slides were examined with an FITC-3540 filter and scanned for at least 5 min before being considered negative (32). DF tests showing typical spirochete morphology and motility and IF tests showing both typical morphology and fluorescent staining were considered positive.

For PCR DNA testing, nucleic acids from genital ulcer swabs or other clinical material were extracted using the Qiagen (Hilden, Germany) blood and tissue DNA extraction kit. Real time-PCR (RT-PCR) was performed using TaqMan universal master mix reagent on an ABI TaqMan 7500 Fast instrument (Applied Biosystems, Thermo Fisher, Waltham, MA, USA). Two syphilis detection targets (TP47 and *polA* genes) and two controls, including a human beta-globin sample adequacy control and an inhibition control containing a known copy number of synthetic DNA, were used. Primers are detailed in Table S1. The PCR test was performed as two duplex reactions in the same run, as follows: 1 cycle of denaturation (95°C for 2 min), followed by 40 cycles of amplification and detection (95°C for 15 s and

60°C for 1 min). The test was considered positive only when both *T. pallidum* targets were detected with a cycle threshold ($C_T$) value of <40. If only one *T. pallidum* target was positive, the inhibition control was ≥2 $C_T$ lower than expected, or the beta-globin control was negative, the specimen was reextracted and tested again.

For PCR DNA and direct detection tests, we defined a sample as a true positive for syphilis based on the following: (i) PCR was positive for both TP47 and *polA* targets; (ii) PCR was positive for one target, and DF and/or IF microscopy was positive; or (iii) PCR was negative, invalid, or not tested, and both DF and IF microscopy were positive together with RPR positivity. For clinical lesion samples that were tested by two or more direct detection tests and were classified as true positives or negatives, we calculated sensitivity, specificity, positive and negative predictive values, and false-positive and -negative rates for each test.

**Azithromycin resistance detection.** Restriction fragment length polymorphism (RFLP) analysis was used to detect point mutations (A2058G and A2059G) associated with macrolide resistance. For all positive PCR tests from clinical samples, the *T. pallidum* 23S rRNA gene was amplified using HotStarTaq master mix (Qiagen, Hilden, Germany) on an ABI9700 thermocycler (Applied Biosystems, Thermo Fisher, MA, USA) as follows: 1 cycle of denaturation (95°C for 15 min), 45 cycles of amplification (95°C for 1 min, 63°C for 2 min, and 72°C for 1 min), and a final elongation step at 72°C for 10 min. The amplicons were stained and visualized on a Gel Doc system (Bio-Rad, Hercules, CA, USA) after 2% agarose gel electrophoresis at 120 V for 1 h. The expected size of the final amplicon is 629 bp. Restriction digestion used two enzyme panels (New England BioLabs, Beverly, MA, USA) at 37°C overnight. Panel 1 consists of MboII and BsaXI targeted to A2058G, while panel 2 consists of BsaI targeted to A2059G. The resulting digested products were separated on an agarose gel, stained, and visualized as described above. Results were interpreted based on the DNA fragment size polymorphism (Table S2).

**Syphilis case counts.** BC syphilis cases from 1 January 2015 to 31 December 2020, stratified by infection stage and reported sexual orientation, were obtained from BCCDC surveillance data.

This study was undertaken as a program evaluation; therefore, ethics approval was not required.

## SUPPLEMENTAL MATERIAL

Supplemental material is available online only.

**SUPPLEMENTAL FILE 1**, PDF file, 0.1 MB.

**SUPPLEMENTAL FILE 2**, PDF file, 0.1 MB.

## ACKNOWLEDGMENTS

We acknowledge the support of the staff of the Zoonotic Diseases and Emerging Pathogens and the High Volume Serology sections of the BCCDC Public Health Laboratory and the staff of the BCCDC Sexually Transmitted Infections clinics.

None of the authors report any conflicts of interest related to this study.

No external funding was received for this work. Siemens Healthcare Diagnostics provided test kits at no charge for the initial laboratory validation that was carried out prior to the implementation of the reverse algorithm.

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
