## [Reviewer comments · Microbiology Spectrum]

Microbiology Spectrum

British Columbia's Experience After Implementation of the *Treponema pallidum* Reverse Algorithm and PCR Detection, 2015 to 2020

Muhammad Morshed, Min-Kuang Lee, Jonathan Laley, Darrel Cook, Annie Mak, Navdeep Chahil, Venessa Ryan, Carolyn Montgomery, Sylvia Makaroff, Sarah Malleson, Barbra Arnold, Troy Grennan, Jason Wong, and Mel Kraiden

Corresponding Author(s): Muhammad Morshed, British Columbia Centre for Disease Control

Review Timeline:

Submission Date:	February 24, 2022
Editorial Decision:	April 21, 2022
Revision Received:	May 9, 2022
Accepted:	May 16, 2022

Editor: Eleanor Powell

Reviewer(s): Disclosure of reviewer identity is with reference to reviewer comments included in decision letter(s). The following individuals involved in review of your submission have agreed to reveal their identity: Daniel A Ortiz (Reviewer #2)

Transaction Report:

DOI: <https://doi.org/10.1128/spectrum.00686-22>

April 21, 2022

Dr. Muhammad G Morshed
British Columbia Centre for Disease Control
Public Health Laboratory
655 West 12th Avenue
Vancouver, BC V5Z 4R4
Canada

Re: Spectrum00686-22 (**British Columbia's Experience After Implementation of the *Treponema pallidum* Reverse Algorithm and PCR Detection, 2015 to 2020**)

Dear Dr. Muhammad G Morshed:

Thank you for submitting your manuscript to Microbiology Spectrum. After receiving feedback from reviewers, modifications will be required before potential publication.

Link Not Available

Sincerely,

Eleanor Powell

Journals Department
Reviewer comments:

Reviewer #1 (Comments for the Author):

Morshed et. al. present a provincial analysis / validation for the implementation of the Syphilis reverse algorithm in British Columbia. The manuscript is well written and would be of interest to your readers. I have some minor comments below that should be addressed before acceptance.

-The authors reference the BCCDC case definition website in the methods. However case definitions are often revised and optimized on a regular basis. For this reason the exact criteria used for syphilis staging should be detailed in the methods

-On a similar theme the introduction could be better served by a brief statement describing the stages of syphilis infection.

-Provide primer sequences for the lab developed PCR test

-The authors mention that penicillin resistance mutations have been identified but only conclude that azithromycin resistance should be monitored. Given that penicillin is the go to treatment, it is vital that surveillance of penicillin resistance also be highlighted given rising rates in Canada.

-Is prenatal screening for syphilis done in BC? If so what are the recommended screening intervals? This was not clear in the manuscript.

Reviewer #2 (Comments for the Author):

Major Comments:

A large retrospective study of the reverse algorithm and PCR-based Tp DNA detection from 2015-2020 was conducted in British Columbia. Azithromycin resistance in DNA positive samples was also examined during the same time period. Authors should explain the two different mutations and how they contribute to azithromycin resistance.

Lines 123-126: Since EIA is also better at detecting primary syphilis, a true syphilis case should also be determined by the following criteria: positive direct detection test (PCR, DF, or IF) and treponemal positivity (EIA and TPPA) or non-treponemal positivity (RPR).

Figure 1: Percentages do not add up for RPR positive (3.5%) and negative (10.6%) under TPPA Negative (10.9%). I believe the 3.5% should be 0.35%.

Line 174: Are these 4,693 EIA positives also TPPA positive? How did the authors determine these were latent cases vs primary vs previously treated or resolved? Were chart reviews conducted? How many were classified as early (confirmed/probable) vs late latent?

Line 178: Were these 814 samples tested by direct detection tests because they were primary cases from chancres?

Lines 194-195: How were congenital cases confirmed? How many congenital cases occurred prior to 2019?

Lines 201-204: With the rise of syphilis cases and the presumptive identification of latent cases, why do we not see more cases of neurosyphilis, ocular syphilis, and tertiary syphilis? My assumption is that most of these latent cases are previously treated or resolved infections. The authors mention that BC maintains a database of all syphilis diagnoses and treatments, but how many were unknowingly treated when taking antibiotics for other infections?

Lines 224-226: Authors should also mention that chancres may not be visible during primary infection if located inside the anal or vaginal cavity.

Lines 226-228: DF also requires a high bacterial burden (around 10⁵ organisms/mL) unlike PCR which has an amplification step that increases sensitivity.

Line 234: The mainstay for syphilis treatment is penicillin. Alternative therapies include doxycycline, tetracycline, and ceftriaxone. Only in rare situations should azithromycin be used so it is unclear why this was included in the study.

Lines 250-252: What were the rates of congenital syphilis prior to 2019-2020?

Minor Comments:

Lines 167-170: Might be good to mention again what is considered a false positive EIA+/TPPA-/RPR-. Also, would be nice to

see number of false positives with percentages in parenthesis for 9.5% and 1.1%. Same for equivocal.

Staff Comments:

Preparing Revision Guidelines

Please return the manuscript within 60 days; if you cannot complete the modification within this time period, please contact me. If you do not wish to modify the manuscript and prefer to submit it to another journal, please notify me of your decision immediately so that the manuscript may be formally withdrawn from consideration by Microbiology Spectrum.

Response to Reviewer Comments

Morshed et al., manuscript number Spectrum00686-22

British Columbia's Experience After Implementation of the *Treponema pallidum* Reverse Algorithm and PCR Detection, 2015 to 2020

The authors thank both reviewers for their constructive comments. Our responses to each point are shown in italics.

Reviewer #1 (Comments for the Author):

Morshed et. al. present a provincial analysis / validation for the implementation of the Syphilis reverse algorithm in British Columbia. The manuscript is well written and would be of interest to your readers. I have some minor comments below that should be addressed before acceptance.

-The authors reference the BCCDC case definition website in the methods. However case definitions are often revised and optimized on a regular basis. For this reason the exact criteria used for syphilis staging should be detailed in the methods

We have included the entire case definition in the supplemental material and updated the reference in the manuscript.

-On a similar theme the introduction could be better served by a brief statement describing the stages of syphilis infection.

We have included a description at the end of the first paragraph of the introduction.

-Provide primer sequences for the lab developed PCR test

We have included the primer sequences in the supplementary material.

-The authors mention that penicillin resistance mutations have been identified but only conclude that azithromycin resistance should be monitored. Given that penicillin is the go to treatment, it is vital that surveillance of penicillin resistance also be highlighted given rising rates in Canada.

We have added to the discussion a description of the reasons for monitoring azithromycin resistance and also commented on the importance of surveillance for penicillin resistance.

-Is prenatal screening for syphilis done in BC? If so what are the recommended screening intervals? This was not clear in the manuscript.

We have added to the discussion that prenatal screening is recommended but not mandatory in BC, and that the uptake exceeds 90%.

Reviewer #2 (Comments for the Author):

Major Comments:

A large retrospective study of the reverse algorithm and PCR-based *Tp* DNA detection from 2015-2020 was conducted in British Columbia. Azithromycin resistance in DNA positive samples was also examined during the same time period. Authors should explain the two different mutations and how they contribute to azithromycin resistance.

*A2058G and A2059G are the *Tp* 23sRNA mutations that have been reported to be associated with macrolide treatment failures. It is unclear to us how their contributions to resistance are relevant to the present analysis. Therefore, we have not addressed this comment.*

Lines 123-126: Since EIA is also better at detecting primary syphilis, a true syphilis case should also be determined by the following criteria: positive direct detection test (PCR, DF, or IF) and treponemal positivity (EIA and TPPA) or non-treponemal positivity (RPR).

We agree but the data were analyzed only on the basis of corresponding RPR results. We have included a statement in the limitations section that not including EIA positive results in this analysis may have resulted in lower calculated sensitivities for the direct tests.

Figure 1: Percentages do not add up for RPR positive (3.5%) and negative (10.6%) under TPPA Negative (10.9%). I believe the 3.5% should be 0.35%.

The reviewer is correct. We have corrected the figure.

Line 174: Are these 4,693 EIA positives also TPPA positive? How did the authors determine

these were latent cases vs primary vs previously treated or resolved? Were chart reviews conducted? How many were classified as early (confirmed/probable) vs late latent?

We have added a statement that these cases underwent chart review. We do not have information on whether they were classified as early vs. late latent; we only had information on whether they received syphilis treatment in the past.

Line 178: Were these 814 samples tested by direct detection tests because they were primary cases from chancres?

Yes, these were suspected syphilis lesions and this has been indicated in the manuscript.

Lines 194-195: How were congenital cases confirmed? How many congenital cases occurred prior to 2019?

Congenital cases were confirmed by laboratory results and clinical assessment. There were no congenital cases identified for the years 2015-2018 (as shown in Table 3).

Lines 201-204: With the rise of syphilis cases and the presumptive identification of latent cases, why do we not see more cases of neurosyphilis, ocular syphilis, and tertiary syphilis? My assumption is that most of these latent cases are previously treated or resolved infections. The authors mention that BC maintains a database of all syphilis diagnoses and treatments, but how many were unknowingly treated when taking antibiotics for other infections?

We agree that both spontaneous resolution and unknowing treatment are the most likely explanations and we have updated the discussion accordingly. We have no data on how many individuals might have received antibiotic treatment for other indications.

Lines 224-226: Authors should also mention that chancres may not be visible during primary infection if located inside the anal or vaginal cavity.

We agree and have added this point to the discussion.

Lines 226-228: DF also requires a high bacterial burden (around 10⁵ organisms/mL) unlike PCR which has an amplification step that increases sensitivity.

While this seems self-evident, we have nevertheless added a statement to the discussion.

Line 234: The mainstay for syphilis treatment is penicillin. Alternative therapies include doxycycline, tetracycline, and ceftriaxone. Only in rare situations should azithromycin be used so it is unclear why this was included in the study.

We have added a statement to the discussion stating the reason for monitoring azithromycin resistance.

Lines 250-252: What were the rates of congenital syphilis prior to 2019-2020?

Table 3 shows that no cases of congenital syphilis were diagnosed from 2015-2018.

Minor Comments:

Lines 167-170: Might be good to mention again what is considered a false positive EIA+/TPPA-/RPR-. Also, would be nice to see number of false positives with percentages in parenthesis for 9.5% and 1.1%. Same for equivocal.

We have added this information. There were some errors in the original numbers cited and these have been corrected.

May 16, 2022

Dr. Muhammad G Morshed
British Columbia Centre for Disease Control
Public Health Laboratory
655 West 12th Avenue
Vancouver, BC V5Z 4R4
Canada

Re: Spectrum00686-22R1 (**British Columbia's Experience After Implementation of the *Treponema pallidum* Reverse Algorithm and PCR Detection, 2015 to 2020**)

Dear Dr. Muhammad G Morshed:

It is my pleasure to tell you that your manuscript has been accepted, and I am forwarding it to the ASM Journals Department for publication. You will be notified when your proofs are ready to be viewed.

Sincerely,

Eleanor Powell
Editor, Microbiology Spectrum

Journals Department
Supplemental Material: Accept
Supplementary Table 1: Accept